# Comparing Communication Methods to Increase Radon Knowledge and Home Testing: A Randomized Controlled Trial in a High-Radon City

**DOI:** 10.3390/ijerph20095634

**Published:** 2023-04-25

**Authors:** Soojung Kim, Hannah Scheffer-Wentz, Marilyn G. Klug, Gary G. Schwartz

**Affiliations:** 1Department of Communication, College of Arts and Sciences, University of North Dakota, Grand Forks, ND 58202, USA; 2Department of Population Health, School of Medicine & Health Sciences, University of North Dakota, Grand Forks, ND 58202, USA

**Keywords:** radon, smartphone, clinical trial, communication, lung cancer, disease prevention

## Abstract

Introduction: Exposure to residential radon is a preventable cause of cancer. Prevention requires testing, but the percentage of homes that have been tested is small. One reason for the low testing rates may be that printed brochures fail to motivate people to obtain and return a radon test. Methods: We developed a radon app for smartphones that contained the same information as printed brochures. We conducted a randomized, controlled trial that compared the app to brochures in a population comprised largely of homeowners. Cognitive endpoints included radon knowledge, attitudes toward testing, perceived severity and susceptibility to radon, and response and self-efficacy. Behavioral endpoints were participants’ requests for a free radon test and the return of the test to the lab. Participants (N = 116) were residents of Grand Forks, North Dakota, a city with one of the nation’s highest radon levels. Data were analyzed by general linear models and logistic regression. Results: Participants in both conditions showed significant increases in radon knowledge (*p* < 0.001), perceived susceptibility (*p* < 0.001), and self-efficacy (*p* = 0.004). There was a significant interaction, with app users showing greater increases. After controlling for income, app users were three times more likely to request a free radon test. However, contrary to expectation, app users were 70% less likely to return it to the lab (*p* < 0.01). Conclusions: Our findings confirm the superiority of smartphones in stimulating radon test requests. We speculate that the advantage of brochures in promoting test returns may be due to their ability to serve as physical reminders.

## 1. Introduction

Radon is an invisible, odorless gas produced by the natural decay of uranium and other radioactive elements present in rocks and soil. Radon enters homes via cracks in the foundation and can accumulate indoors, particularly during cold weather, when homes are tightly sealed. Radon gas undergoes further radioactive decay and, if inhaled, can cause cytotoxic and genotoxic damage to respiratory cells. It is the largest cause of lung cancer among non-smokers and causes more than 21,000 lung cancer deaths in the U.S. per year [1]. Radon may contribute to other lung diseases, including chronic obstructive pulmonary disease (COPD) and childhood asthma [2,3,4,5]. Radon can damage non-respiratory cells present in the respiratory tract, e.g., lymphocytes, and recent epidemiologic studies implicate it in the etiology of stroke [6,7,8]. Thus, radon’s public health importance may extend considerably beyond its established role in lung cancer.

One in 15 U.S. homes has radon levels that exceed the level recommended by the Environmental Protection Agency (EPA) to prevent lung cancer [1]. Despite the Surgeon General’s recommendation that all homes be tested for radon, the vast majority of U.S. housing units (82%) have never been tested [9,10]. There are multiple reasons for the low testing rates, including the public’s limited understanding of radon and the lack of a sense of urgency about imperceptible hazards whose effects have a long latency [11]. 

The low radon testing rates also may reflect a failure of standard communication methods, e.g., printed brochures, to educate people about radon and to encourage them to test their homes. We therefore developed an alternate radon communication method, an application (app) for smartphones. The app contained the same radon information as printed brochures from the EPA but conveyed that information via smartphone. In a randomized trial (N = 138) comparing the app to brochures, the app was markedly superior in increasing radon knowledge and encouraging users to request free radon tests [12]. App users requested a radon test at a rate 3 times that of brochure recipients. However, that trial had important limitations regarding generalizability as it was conducted with a population of college students, a demographic in which homeownership is low and smartphone proficiency is high [13]. 

The goal of this trial was to attempt to replicate the superiority of the radon app in an older, non-college-age population comprised largely of homeowners. We hypothesized that, compared to participants exposed to print brochures, participants exposed to the radon app would show a higher rate of requests for free radon tests and a higher rate of returning them to the laboratory.

## 2. Methods

A randomized, controlled trial design with two arms was used, as described previously [12]. Briefly, participants were randomly assigned to either the experimental condition (the radon app) or usual care (the brochures). Pre-exposure and post-exposure online surveys were constructed on Qualtrics to obtain demographic data and to measure radon knowledge, radon testing attitudes, perceived severity and susceptibility, response efficacy, and self-efficacy.

Participants were parents whose children were enrolled in a daycare facility, supplemented by adults recruited by the parents. The daycare was located in Grand Forks, North Dakota (ND), a city with one of the highest radon levels in the U.S. The average residential radon in Grand Forks is 11.7 pCi/L (433 Bq/m^3^), a value 9 times the U.S. average [14]. The principal eligibility criteria were smartphone ownership and age equal to or greater than 18. The principal exclusion criterion was performing a radon test during the previous two years.

The parents of children who attended the daycare facility received an invitation letter. The letter was placed in the children’s cubbies at the daycare and included a web address and QR code for the online, pre-exposure survey. Only one parent per family was eligible. To ensure adequate enrollment, parents could invite neighbors, friends, and/or co-workers to participate by sharing the link and QR code. Once a potential participant entered the web address or scanned the QR code, he/she was directed to the pre-exposure survey on Qualtrics. Upon accessing the survey, participants were asked to read the informed consent form. If participants proceeded with the online survey, it was considered that they had read the form and had agreed to participate.

After participants answered questions measuring radon knowledge, radon testing attitudes, perceived severity and susceptibility, response efficacy, and self-efficacy, they were assigned to a condition. Participants randomized to the radon app received instructions on how to install the app and were instructed to use it for the next three months. App installation was verified by app activity data saved in a server maintained by the app developer. App users could request a free radon test at any time by using the app. They would receive a charcoal canister test, instructions for using it, and a pre-addressed, pre-paid envelope to use for returning the test to the lab. We used short-term (48–72 h) radon tests (Alpha Energy, Inc., Carrollton, TX, USA). These were bar-coded so that we could count the number of kits ordered and the number returned to the lab.

Brochure condition participants received three EPA brochures during the ensuing three months via U.S. mail. These were: (1) Basic radon facts (https://bit.ly/2S6ryFw (accessed on 1 January 2022)); (2) A citizen’s guide to radon (https://bit.ly/3l3FhcJ (accessed on 1 January 2022)); and (3) Consumer’s guide to radon reduction (https://bit.ly/3ibt83n (accessed on 1 January 2022)). Each mailing also contained a pre-stamped postcard that allowed them to order a free radon test.

All participants answered survey questions regarding demographic data (age, gender, race/ethnicity, housing type, household income, education level, and smoking status). Three months after participants completed the pre-exposure survey, they received an email with a web address for the post-exposure survey. A priori power analyses were performed using PASS (Power Analysis Sample Size Software. 2019, Kaysville, UT, USA) [15]. Calculations utilized an alpha = 0.05 and beta = 0.80. For paired t-tests, we could detect differences of 2 or greater with at least 50 people in each group and differences in proportions of 2% or greater.

Participants received $30 gift cards in exchange for participation in each of the pre-exposure and post-exposure surveys. Parents received a $25 gift card per additional participant they attracted, for a maximum of 3 per parent. This study was approved by the IRB at the University of North Dakota (IRB # IRB0004359) and was registered in ClinicalTrials.gov (ID: NCT05319457).

The cognitive endpoints were similar to those we described previously [12]. Briefly, we measured radon knowledge as the number of accurate responses to 20 True–False statements about radon. Attitudes about testing were measured by five 7-point semantic differential scales (e.g., “bad–good”). Perceived severity is the extent to which an individual believes that the threat (radon) will result in negative health outcomes, and perceived susceptibility is the extent to which an individual believes that he/she is personally vulnerable to that threat. Response efficacy is the individuals’ evaluation of the effectiveness of recommended health behaviors. Finally, self-efficacy is an individual’s perceived capability to perform the recommended behaviors (e.g., “I feel like I could easily perform a radon test”). Each of these variables was measured by two 7-point Likert scales. These measurement scales have been used in previous studies and were shown to be valid and reliable [12,16,17,18,19].

There were two behavioral outcomes: the rate at which participants ordered a free radon test kit and the rate at which they returned it to the testing lab. Both were measured via bar codes on the tests. 

To create radon knowledge index variables, a correct response was coded as “1” and an incorrect response, or “don’t know,” as “0.” We created two radon knowledge index variables for T1 and T2 whose sums reflect correct responses to 20 True–False radon knowledge statements. Two radon-testing attitudes variables were created by averaging five semantic differential scales in T1 and T2. These showed excellent reliability (Cronbach’s α in T1 = 0.95; Cronbach’s α in T2 = 0.90). Perceived severity variables were created using two Likert scales in T1 (Spearman–Brown coefficient = 0.91) and T2 (Spearman–Brown coefficient = 0.79). Perceived susceptibility, response efficacy, and self-efficacy variables were created similarly (Spearman–Brown coefficient for these variables were ≥0.80). 

The characteristics of participants in the two conditions were compared by Chi-Square. Changes in the cognitive variables were compared via paired t-tests. General linear models with repeated measures with interaction were used to detect differences in changes over time between the conditions. Differences in the proportion of participants in the two conditions who ordered a free radon test kit and those who returned it to the lab were compared using Z tests. Logistic regressions were used to test these differences while accounting for participant characteristics. Relative risks were estimated using stratified 2 × 2 tables.

## 3. Results

The CONSORT diagram for the trial is shown in Figure 1. Enrollment began in May 2022. There were 116 potential participants, 39 were parents and 77 were individuals referred by them. Twelve (12) potential participants (10.3%) were ineligible because they had tested their houses in the previous two years. The remaining 104 participants were randomized. Twenty (20) of the participants randomized (19.2%) were lost to follow-up, 8 in the app condition and 12 in the brochure condition, leaving 84 participants in T2, 44 in the app and 40 in the brochure condition. 

Characteristics of participants are shown in Table 1. The average age was 38.95, ranging from 19 to 79 years (SD = 13.24). Seventy-five (75) participants (72.1%) were female and 29 (27.9%) were male. The majority were white (94.2%) and were homeowners (70.2%). Approximately half reported a household income of less than $100,000. Most (56.73%) held bachelor’s or post-graduate degrees. Tobacco use was reported by six participants (5.8%), and e-cigarette use by three (2.9%). 

After randomization, there were no significant differences in both groups with respect to sex, age, race/ethnicity, education level, housing type, and tobacco and/or e-cigarette use (Table 1). However, participants in the brochure group had a significantly higher income (*p* = 0.028). For example, nearly twice as many individuals in the brochure group (56%) had an annual income over $100,000 vs. 29% of the app participants.

Figure 2 shows the average scores pre- and post-exposure to condition in app users and brochure recipients. Radon knowledge showed the greatest change: an increase of seven points for the radon app condition vs. six for the brochure condition (*p* < 0.001). App users showed significantly more positive radon testing attitudes (*p* = 0.040), higher perceived susceptibility (*p* < 0.001), and higher self-efficacy (*p* < 0.001) than brochure recipients. Brochure recipients had a significant increase in their self-efficacy scores (*p* = 0.023). Radon knowledge (*p* < 0.001), perceived susceptibility (*p* < 0.001), and self-efficacy (*p* = 0.004) showed significant interactions. 

We hypothesized that requests for free radon kits and the return of the kits to the lab would be higher in the radon app condition. Thirty-six (36) of the 52 participants assigned to the radon app condition (69%) ordered a kit, vs. 28 of the 52 participants assigned to the brochure condition (54%) (*Z* = 1.612, *p* = 0.053). However, 14 of the 36 (39%) participants in the radon app condition who ordered the test kit returned it to the lab, vs. 19 of the 28 (68%) participants in the brochure condition who ordered the test kit returned it to the lab (*Z* = 2.300, *p* = 0.011) (Figure 3). 

In univariate models, homeownership was significantly associated with ordering a test kit. However, when examined via logistic regression, the only participant attribute significantly related to conditions was income (see Table 2). When income was added to the model, both the radon app and income were strongly associated with test ordering. Participants using the app were three times more likely to order a test kit (OR = 3.46). Individuals with the lowest income were 80% less likely to order a test (OR = 0.20). Conversely, for 64 participants who had ordered a test kit, those using the app were 70% less likely to return it (OR = 0.30).

## 4. Discussion 

This randomized trial sought to replicate our previous findings of the superiority of the radon app on radon knowledge and behaviors over printed brochures in a broad sample of the population in Grand Forks, ND, a city with one of the highest residential radon levels in the U.S. Our most important findings are that, after adjustment for income, the app markedly outperformed brochures in stimulating requests for radon tests, a finding consistent with our previous findings. However, contrary to expectations, the app was significantly inferior to brochures in promoting individuals to return the tests to the lab. Together, these findings provide actionable information for future efforts to prevent radon-induced disease.

Both the radon app and brochures were effective in improving radon knowledge (*p* < 0.001), with a slight advantage for app users over brochure recipients (i.e., an improvement of 7 vs. 6 points in radon knowledge, respectively). Similarly, app users demonstrated a significantly higher perceived susceptibility to radon as well as more positive attitudes toward radon testing, both of which are consistent with the superiority of the app as an educational method. Importantly, app users requested radon tests at a higher rate than brochure recipients (69% vs. 54%). Although this difference was marginally significant in univariate testing (*p* = 0.053), the performance of the app in stimulating test requests was significantly better than the brochures in logistic models that adjusted for income. Specifically, after income adjustment, app users were three times more likely to order free test kits. This finding is identical in magnitude to our previous findings with a college-student population [12].

We found that individuals with the lowest income were 80% less likely to order free tests, a finding consistent with the sizable literature on radon testing, e.g., [11,20]. Although income often is considered to be a surrogate for education, income per se may influence test requests since individuals who cannot afford to remediate their homes for radon may be less motivated to test for it. Although homeownership was a significant predictor of test requests in univariate analyses, only income and condition were significant predictors of testing in the multivariate, logistic model. We note that, despite randomization, income was significantly higher in the brochure condition. Adjusting for income thus was essential to produce an unbiased measure of the effects of the two conditions. In future studies, it may be valuable to stratify by income before randomization to ensure its equal distribution.

After adjustment for income, brochure recipients were significantly less likely to request test kits. However, brochure recipients who requested kits were almost twice as likely as app users to return them to the lab (68% vs. 39%). Although based on relatively small numbers (19/28 vs. 14/36, brochure and app, respectively), this finding was contrary to expectation. The discrepancy between requesting a test and returning the test underscores the fact that the question, “What are the determinants of radon testing rates?”, is a complex one that encompasses several different questions and behaviors. That is, in order to test for radon, individuals first must learn about radon, then acquire a test, and then return the test to the lab. Regarding the first step, learning about radon, the app was equivalent or slightly superior to brochures (as evidenced by the greater increase in radon knowledge in the app group). With regard to the second step, acquiring a test, the app was clearly superior. Conversely, concerning the third step, returning the test to the lab, the app was inferior. How can this be understood? 

We suggest that the processes involved in test acquisition and test return are different. The superiority of the app in the initial two steps appears to be due to its superiority in engaging users. This is evidenced by the difference in loss to follow-up in the two conditions. In the present and our previous trial, loss to follow-up after randomization was approximately 50% less in the app than in the brochure condition (15 vs. 23% and 19 vs. 26%, app vs brochure, respectively) [12]. This suggests that participants found the app more engaging than the brochures. 

Conversely, behaviors relevant to returning a radon test may be influenced by factors other than engagement. Unlike the app, the brochures have an actual physical presence. We speculate that the brochures may have provided a tangible reminder to follow through. It is noteworthy that physical reminders, e.g., postcards, have been shown to increase test return rates in an analogous setting, i.e., the use of at-home, fecal occult blood tests for colorectal cancer [21,22].

Our study has several limitations. First, 19% of participants were lost to follow-up, which could introduce some selection bias. This rate is comparable to average losses to follow-up in other trials of behavior change (i.e., 18%) [23]. Secondly, our participants had higher education and income than the U.S. population overall, as they were recruited from clients utilizing a daycare facility (i.e., they were likely to be employed) [24]. Nonetheless, we considered adults in Grand Forks, ND to be an appropriate sample because the radon levels in Grand Forks are among the highest in the U.S. [25,26,27]. Approximately 10% of potential participants had previously tested their homes, a value approximately similar to the percentage of homes tested nationwide (18%). The value is lower than the U.S. overall, likely because, unlike most states, ND has no laws requiring radon testing at the time of home sales [28]. Although our study included a higher percentage of non-Hispanic Whites than the general U.S. population, its composition accurately reflects the racial composition of the Midwest [29]. Lastly, our finding that the brochures were superior to the app in stimulating test returns was contrary to expectation and requires confirmation.

Conversely, our study has several strengths: it was randomized, adequately powered, and compared a novel intervention (the radon app) to usual care (brochures) with a sample comprised largely of homeowners in a high-radon city. Moreover, unlike the majority of studies regarding radon testing, which measure subjective attitudes and/or “readiness” to test, we measured actual test acquisition and test returns using objective measures (bar-coded tests) that are independent of self-report [30]. The consistency of our results with our previous findings in a college-aged population suggests that the superiority of the radon app on cognitive outcomes and in stimulating requests for radon tests is genuine.

## 5. Conclusions

The results of this randomized controlled trial confirmed the superiority of the radon app over printed brochures in numerous aspects of radon education and especially, in stimulating individuals to obtain a radon test kit, the first step in radon testing. Conversely, our prediction that the app would be more effective than brochures in prompting individuals to return the test to the lab was not confirmed. We speculate that the superiority of the printed brochures in prompting test returns is due to their ability to serve as physical reminders. Future studies aimed at maximizing the efficacy of the radon app in stimulating radon testing should test the effect of physical reminders, e.g., postcards, on test return rates. 

## Figures and Tables

**Figure 1 ijerph-20-05634-f001:**
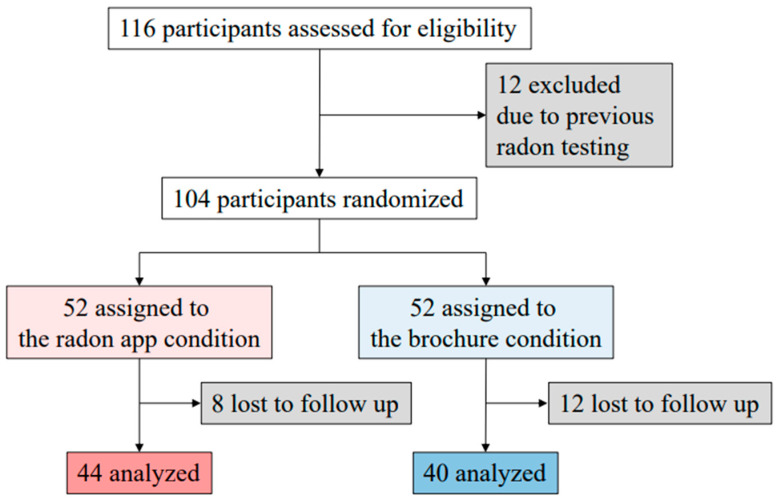
Consort diagram of clinical trial.

**Figure 2 ijerph-20-05634-f002:**
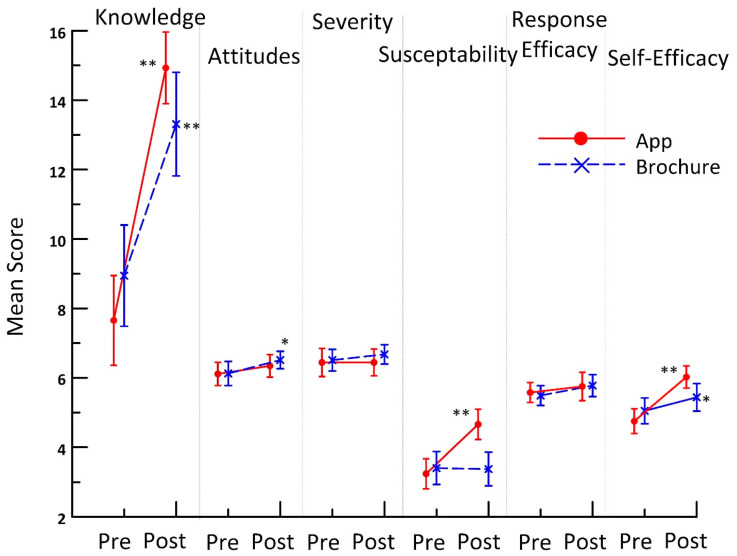
Scores on radon knowledge and attitudes pre- and post-exposure to radon information delivered via brochure and via smartphone. * Indicates *p* < 0.05 and ** indicates *p* < 0.01.

**Figure 3 ijerph-20-05634-f003:**
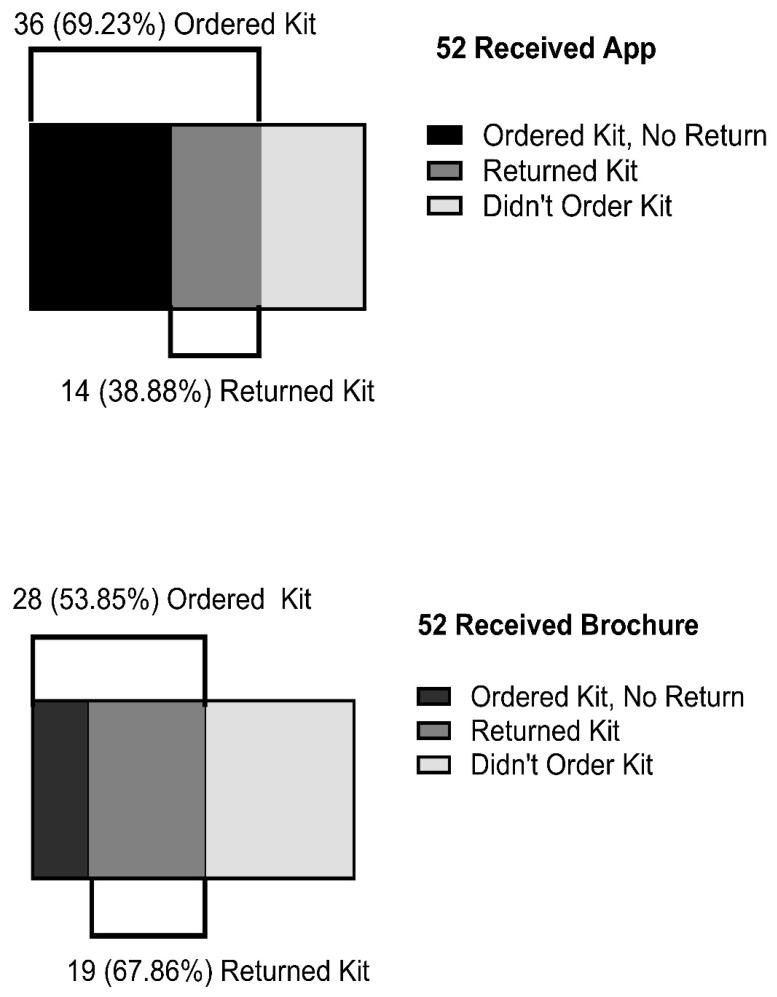
Number and percent of individuals ordering and returning radon test kits by experimental condition, smartphone app vs. brochure.

**Table 1 ijerph-20-05634-t001:** Participant characteristics by condition.

		All		Radon App		Brochure	
		N	%	N	%	N	%
Sex	Male	29	27.88	13	25.00	16	30.77
	Female	75	72.12	39	75.00	36	69.23
Age	19–28	26	25.00	15	28.85	11	21.15
	29–38	29	27.88	16	30.77	13	25.00
	39–48	25	24.04	9	17.31	16	30.77
	48+	24	23.08	12	23.08	12	23.08
Race	White	98	94.23	51	98.08	47	90.38
	Other	6	5.77	1	1.92	5	9.62
Housing	Owner	73	70.19	34	65.38	39	75.00
	Rental	25	24.04	15	28.85	10	19.23
	Other	6	5.77	3	5.77	3	5.77
Income	<$50,000	24	23.08	15	28.85	9	17.31
	$50,000–$100,000	30	28.85	18	34.62	12	23.08
	>$100,000	44	42.31	15	28.85	29	55.77
	Unknown	6	5.77	4	7.69	2	3.85
Education	<Post-high school	25	24.04	13	25.00	12	23.08
	College graduate	39	37.5	21	40.38	18	34.62
	Post-graduate	20	19.23	10	19.23	10	19.23
	Unknown	20	19.23	8	15.38	12	23.08
Tobacco	Current	6	5.77	2	3.85	4	7.69
	None	98	94.23	50	96.15	48	92.31
E-Cigs	Current	3	2.88	1	1.92	2	3.85
	None	101	97.12	51	98.08	50	96.15

**Table 2 ijerph-20-05634-t002:** Incidence of test kit ordering between radon app and brochure conditions by income.

	Beta	SE B	*P*	OR	95% CI	
					LL	UL
Ordering a kit without income (n = 104)						
Intercept	−0.32	0.19				
App	0.66	0.41	0.109	1.93	0.86	4.30
Ordering a kit with income (n = 98)						
Intercept	0.10	0.30				
App	1.24	0.49	0.011 *	3.46	1.32	9.05
Income < $50,000	−1.61	0.60	0.007 *	0.20	0.06	0.64
Income $50,000–$100,000	−0.79	0.55	0.149	0.45	0.15	1.33
Testing the house without income (n = 64)						
Intercept	0.68	0.32				
App	−1.20	0.53	0.024 *	0.30	0.11	0.85
Testing the house with income (n = 62)						
Intercept	1.06	0.42				
App	−1.13	0.58	0.053	0.32	0.10	1.01
Income < $50,000	−1.95	0.89	0.028 *	0.14	0.02	0.81
Income $50,000–$100,000	−0.29	0.62	0.637	0.75	0.22	2.53

* *p* < 0.05. Reference value for income was >$100,000.

## Data Availability

The original data for this trial are not publicly available but could be made available upon reasonable request to the corresponding author.

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
