# Peer review of "Comparing Communication Methods to Increase Radon Knowledge and Home Testing: A Randomized Controlled Trial in a High-Radon City"

_ijerph, 2023, doi:10.3390/ijerph20095634_

Round 1

Reviewer 1 Report

  The manuscript "Comparing communication methods to increase radon knowledge and home testing: a randomized controlled trial in a high-radon city" by Soojung Kim, Hannah Scheffer-Wentz, Marilyn G. Klug and Gary G. Schwartz deals with the indoor radon problem.   The impact of radon on human health is still unsolved and underestimated. Still, we know it can cause cancer, but it is hard to measure in people's houses.   The paper is well-written and organized, and the analyses are well-planned and done. A lot of data were taken into account. The introduction is sufficient and gives the main information. The methods have also been well described. The trials were randomized, and the population was presented in detail, such as sex, age, race, income, etc., including cigarettes. The whole framework was presented.   Besides the study's strengths, the authors have also mentioned its limitations. It confirms their unbiased and open way of the study. The paper joins radioactivity analysis and sociological aspects - this is always hard to combine. However, the authors try to find the reasons and effects of people's actions in the case of radon measurements. The authors widely discuss the potential impact on population decisions in the case of taking care in the radiation protection areas. This is a good aspect of the paper.   SI unit for activity concentration is Bq/L, so the authors should change "pCi/L" to Bq/L or Bq/m3.   As we all know, radioactivity can be dangerous; I encourage the authors to mention Cohen's studies on radon in the USA and discuss his thesis.

Author Response

Reviewer 1 commented that “The paper is well-written and organized, and the analyses are well-planned and done”.  The reviewer requests that radioactivity be expressed in SI units, in addition to the pCi/L (which is the common unit in the United States).  We have reported the radon in Grand Forks in both units on Lines 73-74 (“The average residential radon in Grand Forks is 11.7 pCi/L (433 Bq/m3), a value nine times the U.S. average”.)

The reviewer also mentions the possibility of commenting on the work of Cohen in the USA.  As readers may know, Dr. Bernard Cohen was a physicist who (among other things), held that small does of radon might prevent lung cancer.  We have commented on that idea previously (Schwartz et al., Future Oncology 2016) in a paper cited as Ref 25, and argued that it was an error. We do not think it requires recitation here, as that view has generally been discredited (Heath CW Jr, et al. Residential radon exposure and lung cancer risk: commentary on Cohen’s county-based study. Health Physics 2004;87:657-655.)

Reviewer 2 Report

The aim of this work is to attempt to replicate the superiority of the radon app in an older, non-college-age, population comprised largely of homeowners. Authors hypothesized that compared to participants exposed to print brochures, participants exposed to the radon app would show a higher rate of requests for free radon tests and a higher rate of returning them to the laboratory.

They conclude that 

The results of this randomized trial confirm the superiority of the radon app over printed brochures in numerous aspects of radon education and especially, in stimulating individuals to obtain a radon test kit, the first step in radon testing. 

There is no doubt about radon problems all over the world, So the Radon topic is interesting everywhere.  

The idea is shiny in added value to the radiation protection society.

Results were presented well and discussed in an informative way.

 It is suitable to publish in Journal IJERPH with some minor comments

-Please present your goal in a septet paragraph with more stress about work novelty.

- Please use passive in text, no need to say each time WE 

Author Response

Reviewer 2 noted that the paper provides “added value to the radiation protection society” and that the “results were presented well and discussed in an informative way”.  The review asks that that the we present the goal in a separate paragraph and to use the passive voice in the text.

We have now made the goal of the study an independent paragraph, as requested, which appears on Line 59.  The request to use the passive voice may reflect different scientific guidelines in different countries as writers in the United States are generally encouraged to use the active, rather than the passive voice.  Nonetheless, in response to the Reviewers’ request, we have changed many of the “we” statements (e.g., “we used a randomized controlled design”) to the passive voice, “a randomized controlled design was used”, to avoid monotony.

Reviewer 3 Report

This work discusses factors that (can) affect the attitudes of households related to radon levels inside their homes.

It is complete and well written. Methodology and results are well presented and discussed to generate relevant conclusions. Despite that, I will leave some possible small mistakes and some more general questions to authors:

1-   Line 152: 39 plus 79 is not 116;

2-   Line 190: is missing to refer to Figure 3;

3-   Line 193: can omit “see”;

4-   Line 198: Can refer again to Table2;

5-   Line 261: remainders with app would be physical post cards (etc) or a virtual remainder in the app!?;

6-   Line 248: it is a good question. Can this sample size be low to answer that question=(104 or 116)?!

7-   Line 257: The physical presence would influence the third step but not the second one! And about the physical presence of the kit? Could you specify in line 93 the number of days that this kit should be “sampling”.

8-   As specified in line 44, mostly in critical areas/cities, all houses should be tested for radon. In that sense, and to approach that limit, conclusions could indicate that public authorities need to be more instructive with low income homeowners that tend to care less for this issues (as poor people have other more urgent necessities).

Author Response

Reviewer 3 notes that the paper “is complete and well written. Methodology and results are well presented and discussed to generate relevant conclusions”.  The Reviewer also notes “several possible small mistakes and some general questions to authors”. The mistakes include:

  1. Line 152: 39 plus 79 is not 116.

We thank the Reviewer for catching this typo.  The 79 is an error—we have changed this to the correct number, 77.

  1. Line 190—missing to refer to Figure 3. We have inserted a “call out” to Figure 3.
  2. Line 193, omit “see”.

The Reviewer asks whether the sample size might be too low to answer the question, why the app was inferior to the brochures in stimulating test returns? It is indeed possible that this difference may be influenced by the sample size, and we now note that possibility in the Discussion.  However, we believe that, especially since this finding was the reverse of our prediction, it is possible that it is real (not merely a chance effect) and therefore we offered some speculation as to why it might have occurred.

We agree with the Reviewer that “public authorities need to be more instructive with low income homeowners” with regard to radon.  Unfortunately, North Dakota has no laws at all requiring radon testing, and one of our goals is to draw attention to this important omission which differentially affects the less affluent and less educated.

Reviewer 4 Report

The trial is interesting because it deals with the relevant problem of radon air pollution in Grand Forks (North Dakota). In order to increase peoples awareness of residential radon related issues, print brochures and an app for smartphones have been used with the aim to encourage them to perform free residential radon tests.

Although I have no doubts about the usefulness of the trial, but I have some observations:

     1.          It would be appropriate to identify the purpose of this trial.

     2.        There is a lack of justification as to why, for instance, app users requested a free radon test at a rate 3 times that of brochure recipients or as to why  the participants exposed to print brochures showed a higher rate of returning tests to the laboratory. What are the reasons for such choice of radon tests at home?

     3.       The references should be arranged according to the requirements set by the journal. Now full and abbreviated titles of the journals are provided interchangeably.

     4.       It would be appropriate to continue and expand the trial by including the combined analysis of the results obtained from the survey of trial participants about the incidence of any lung diseases and the residential radon tests.

Author Response

Reviewer 4 notes that “there is a lack of justification as to why, for instance, app users requested a free radon test at a rate 3 times that of brochure recipients or as to why, the participants exposed to print brochures showed a higher rate of returning tests to the laboratory.”

We interpret the Reviewer’s comments about “justification” to mean “explanation”.  As we have tried to show in this, and in the previous clinical trial, whose results we have replicated, the app is a more engaging tool than brochures and this is the likely reason it outperformed the brochures in stimulating test requests.  We noted that it also outperformed the brochures in many of the cognitive endpoints and was more successful than the brochures in maintaining participation in the trial, as the brochure condition had a much higher rate of loss to follow-up.

The Reviewer notes some inconsistencies in the Reference section.  The references have now been aligned with IJERPH formatting.

We agree with the Reviewer that it would indeed be interesting to follow-up the participants for incidences of lung disease.  However, that step would require a prospective study that is far beyond our abilities and goals in the present trial.